Software engineering principles to improve quality and performance of R software

Russell Seth 1 seth.russell@ucdenver.edu
http://orcid.org/0000-0003-1483-4236 Bennett Tellen D. 1 2
http://orcid.org/0000-0001-6618-1316 Ghosh Debashis 1 3
1 University of Colorado Data Science to Patient Value, University of Colorado Anschutz Medical Campus , Aurora, CO , USA
2 Pediatric Critical Care, University of Colorado School of Medicine , Aurora, CO , USA
3 Department of Biostatistics and Informatics, Colorado School of Public Health , Aurora, CO , USA
Ventura Sebastian
Electronic publication date: 2019 Feb 4
Publication date: 2019
Volume: 5
Electronic Location ID: e175
Received 2018 Oct 12; Accepted 2019 Jan 11
Copyright: © 2019 Russell et al.
Copyright year: 2019
Copyright holder: Russell et al.
License: This is an open access article distributed under the terms of the Creative Commons Attribution License, which permits unrestricted use, distribution, reproduction and adaptation in any medium and for any purpose provided that it is properly attributed. For attribution, the original author(s), title, publication source (PeerJ Computer Science) and either DOI or URL of the article must be cited.
License URL: https://creativecommons.org/licenses/by/4.0/

Keywords: Unit testing, Profiling, Optimization, Software engineering, R language, Statistical computing, Case study, Reproducible research, Data science

Funding: University of Colorado Data Science to Patient Value The University of Colorado Data Science to Patient Value initiative provided funding for this work. The funders had no role in study design, data collection and analysis, decision to publish, or preparation of the manuscript.

==============================
Today’s computational researchers are expected to be highly proficient in using software to solve a wide range of problems ranging from processing large datasets to developing personalized treatment strategies from a growing range of options. Researchers are well versed in their own field, but may lack formal training and appropriate mentorship in software engineering principles. Two major themes not covered in most university coursework nor current literature are software testing and software optimization. Through a survey of all currently available Comprehensive R Archive Network packages, we show that reproducible and replicable software tests are frequently not available and that many packages do not appear to employ software performance and optimization tools and techniques. Through use of examples from an existing R package, we demonstrate powerful testing and optimization techniques that can improve the quality of any researcher’s software.

Introduction

Writing scientific software has progressed from the work of early pioneers to a range of computer professionals, computational researchers, and self-taught individuals. The educational discipline of computer science, standardized many years ago through recommendations from the Association for Computing Machinery (ACM) (Atchison et al., 1968), has grown in breadth and depth over many years. Software engineering, a discipline within computer science, “seeks to develop and use systematic models and reliable techniques to produce high-quality software. These software engineering concerns extend from theory and principles to development practices that are most visible to those outside the discipline” (The Joint Task Force on Computing Curricula, 2015).

As they gain sophistication, computational researchers, statisticians, and similar professionals need to advance their skills by adopting principles of software engineering. Wilson et al. (2014) identified eight key areas where scientists can benefit from software engineering best practices. The term “best” as referenced in the previous cited work and others cited later refers to expert consensus based on knowledge and observational reporting of results from application of the practices. They provide a high-level description of eight important principles of software engineering that should “reduce the number of errors in scientific software, make it easier to reuse, and save the authors of the software time and effort that can used for focusing on the underlying scientific questions.” While their principles are still relevant and important today, there has not been enough progress in this endeavor, especially with respect to software testing and software optimization principles (Wilson, 2016; Nolan & Padilla-Parra, 2017).

The ACM/Institute of Electrical and Electronics Engineers recommendations for an undergraduate degree in software engineering describe a range of coursework and learning objectives. Their guidelines call out 10 specific knowledge areas that should be part of or guide all software engineering coursework. The major areas are: computer science fundamentals, math and engineering fundamentals, professional interactions/communication, software modeling, requirement gathering, software design, verification, project processes, quality, and security (The Joint Task Force on Computing Curricula, 2015). These major themes are not covered extensively outside software engineering and include such generally applicable items such as software verification, validation, testing, and computer science fundamentals (for example, software optimization, modeling, and requirement gathering).

In addition to the need for further training, understanding the software lifecycle is necessary: the process of software development from ideation to delivery of code. The largest component of software’s lifecycle is maintenance. Software maintenance costs are large and increasing (Glass, 2001; Dehaghani & Hajrahimi, 2013; Koskinen, 2015); some put maintenance at 90% of total software cost. The chief factor in software maintenance cost is the time of the people creating and using the software. From the recent trend on making research results reproducible and replicable, some recommend making code openly available to any who might wish to repeat or further analyze results (Leek & Peng, 2015). With the development of any software artifact, an important consideration for implementation should be maintenance. As research scientists tend to think of their software products as unique tools that will not be used regularly or for a long period, they often do not consider long term maintenance issues during the development phase (Sandve et al., 2013; Prins et al., 2015). While a rigorous and formal software engineering approach is not well suited to the standard lifecycle of research software (Wilson, 2016), there are many techniques that can help to reduce cost of maintenance and speed development. While best practices such as the use of version control software, open access to data, software, and results are becoming more wide spread, other practices such as testing and optimization need further attention.

In this paper, a brief survey of currently available R packages from The Comprehensive R Archive Network (CRAN) will be used to show the continued need for software testing and optimization. Source code for this analysis is freely available at https://github.com/magic-lantern/SoftwareEngineeringPrinciples. After the presentation of the current state of R packages, general advice on software testing and optimization will be presented. The R package “pccc: Pediatric Complex Chronic Conditions” (Feinstein et al., 2018; DeWitt et al., 2017) (pccc), available via CRAN and at https://github.com/CUD2V/pccc, is used for code examples in this article. pccc is a combined R and C++ implementation of the Pediatric Complex Chronic Conditions software released as part of a series of research papers (Feudtner, Christakis & Connell, 2000; Feudtner et al., 2014). pccc takes as input a data set containing International Statistical Classification of Diseases and Related Health Problems (ICD) Ninth revision or Tenth revision diagnosis and procedure codes and outputs which if any complex chronic conditions a patient has.

Analysis of R Packages on Cran

Testing of R packages

In order to estimate the level of testing common among R software, we analyzed all R packages available through CRAN. Although Nolan & Padilla-Parra (2017) performed a similar analysis in the past, due to the rapid change in the CRAN as a whole, a reevaluation is necessary. At the time of Nolan’s work, CRAN contained 10,084 packages; it now contains 13,509. Furthermore, the analysis by Nolan had a few shortcomings that we have addressed in this analysis: there are additional testing frameworks for which we wanted to analyze their usage; not all testing frameworks and R packages store their test code in a directory named “tests”; only packages modified in the past 2 years were reported—there are many commonly used R packages that have not been updated in the last 2 years.

Although we address some shortcomings in analyzing R code for use of testing best practices, our choice of domain for analysis does have some limitations. Not all research software is written in R; for those that do use R, not all software development results in a package published on CRAN. While other software languages have tools for testing, additional research would be needed to evaluate level of testing in those languages to see how it compares to this analysis. Although care has been taken to identify standard testing use cases and practices for R, testing can be performed in-line through use of core functions such as stop() or stopifnot(). Also, developers may have their own test cases they run while developing their software, but did not include them in the package made available on CRAN. Unit tests can be considered executable documentation, a key method of conveying how to use software correctly (Reese, 2018). Published research that involves software is not as easy to access and evaluate for use of testing code as CRAN packages are. While some journals have standardized means for storing and sharing code, many leave the storing and sharing of code up to the individual author, creating an environment where code analysis would require significant manual effort.

To analyze use of software testing techniques, we evaluated all CRAN packages on two different metrics: Metric 1: In the source code of each package, search for non-empty testing directories using the regular expression pattern “[Tt]est[^/]*/.+”. All commonly used R testing packages (those identified for metric 2) recommend placing tests in a directory by themselves, which we look for.

Metric 2: Check for stated dependencies on one of the following testing packages: RUnit (Burger, Juenemann & Koenig, 2015), svUnit (Grosjean, 2014), testit (Xie, 2018), testthat (Wickham, 2011), unitizer (Gaslam, 2017), or unittest (Lentin & Hennessey, 2017). From the authors of these packages, it is recommended to list dependency (or dependencies) to a testing framework even though standard usage of a package may not require it.

For the testing analysis, we used 2008 as the cutoff year for visualizations due to the low number of packages last updated prior to 2008.

As shown in Fig. 1, the evaluation for the presence of a non-empty testing directory shows that there is an increasing trend in testing R packages, with 44% of packages updated in 2018 having some tests. Table S1 contains the data used to generate Fig. 1.

Figure 1 Packages with non-empty testing directory.

Count of packages with files in standard testing directories by year a package was last updated. Testing directory “Yes” is determined by the presence of files matching the regular expression “[Tt]est[^/]*/.+”; if no matches are found for an R package, it is counted as a “No”.

As shown in Fig. 2, reliance upon testing frameworks is increasing over time both in count and as a percentage of all packages. There 16 packages that list dependencies on more than one testing framework (nine with dependencies on both RUnit and testthat, seven with dependencies on both testit and testthat), so the total number of packages shown in the histogram includes 16 that are double counted. Table S2 contains the data used to generate Fig. 2.

Figure 2 Packages with testing framework dependency.

Count of dependencies on a testing package (RUnit, svUnit, testit, testthat, unitizer, unittest) by year a package was last updated. Packages with no stated dependency from their DESCRIPTION file for one of the specified packages are listed as “none”.

As the numbers from Metric 1 do not match the numbers of Metric 2, some additional exploration is necessary. There are 884 more packages identified from Metric 1 vs Metric 2. There are 1,115 packages that do not list a dependency to a testing framework, but have a testing directory; for example, the package xlsx (Dragulescu & Arendt, 2018). Some packages use a testing framework, but do not list it as a dependency; for example, the package redcapAPI (Nutter & Lane, 2018). There are also 231 packages that list a testing framework as a dependency, but do not contain a directory with tests. See Tables S1 and S2 for more details.

Optimization of R packages

In order to estimate the level of software optimization common among R software, we performed an analysis of all R packages available through CRAN. To analyze the use of software optimization tools and techniques, we evaluated all CRAN packages on two different metrics: Metric 1: In the source code of each package, search for non-empty src directories using the regular expression pattern “src[^/]*/.+”. By convention, packages using compiled code (C, C++, Fortran) place those files in a “/src” directory.

Metric 2: Check for stated dependencies on packages that can optimize, scale performance, or evaluate performance of a package. Packages included in analysis are: DSL (Feinerer, Theussl & Buchta, 2015), Rcpp (Eddelbuettel & Balamuta, 2017), RcppParallel (Allaire et al., 2018a), Rmpi (Yu, 2002), SparkR (Apache Software Foundation, 2018), batchtools (Bischl et al., 2015), bench (Hester, 2018), benchr (Klevtsov, Antonov & Upravitelev, 2018), doMC (Calaway, Analytics & Weston, 2017), doMPI (Weston, 2017), doParallel (Calaway et al., 2018), doSNOW (Calaway, Corporation & Weston, 2017), foreach (Calaway, Microsoft & Weston, 2017), future (Bengtsson, 2018), future.apply (Bengtsson & R Core Team, 2018), microbenchmark (Mersmann, 2018), parallel (R Core Team, 2018), parallelDist (Eckert, 2018), parallelMap (Bischl & Lang, 2015), partools (Matloff, 2016), profr (Wickham, 2014a), profvis (Chang & Luraschi, 2018), rbenchmark (Kusnierczyk, Eddelbuettel & Hasselman, 2012), snow (Tierney et al., 2018), sparklyr (Luraschi et al., 2018), tictoc (Izrailev, 2014).

For the optimization analysis, we used 2008 as the cutoff year for visualizations showing presence of a src directory due to the low number of currently available packages last updated prior to 2008. For optimization related dependencies, in order to aid visual understanding, we used 2009 as the cutoff year and only showed those packages with 15 or greater dependent packages in a given year.

Automatically analyzing software for evidence of optimization has similar difficulties to those mentioned previously related to automatically detecting the use of software testing techniques and tools. The best evidence of software optimization would be in the history of commits, unit tests that time functionality, and package bug reports. While all R packages have source code available, not all have development history available nor unit tests available. Additionally, a stated dependency on one of the optimization packages listed could mean the package creators recommend using that along with their package, not that they are actually using it in their package. Despite these shortcomings, it is estimated that presence of a src directory or the use of specific packages is an indication that some optimization effort was put into a package.

As shown in Fig. 3, the evaluation for the presence of a non-empty src directory shows that there is an increasing trend in using compiled code in R packages, by count. However, when evaluated as a percent of all R packages, the change has only been a slight increase over the last few years. Table S3 contains the data used to generate Fig. 3.

Figure 3 Packages with non-empty src directory.

Count of packages with files in standard source directories that has code to be compiled by year a package was last updated. Compiled directory “Yes” is determined by the presence of files matching the regular expression “src[^/]*/.+”; if no matches are found for an R package, it is counted as a “No”.

As shown in Fig. 4, in 2018, Rcpp is the most common optimization related dependency followed by parallel and foreach. Those same packages have been the most popular for packages last updated during the entire period shown. There 699 packages that list dependencies to more than one optimization framework (407 with 2 dependencies, 220 w/3, 53 w/4, 16 w/5, 2 w/6, 1 w/7), so the total number of packages shown in the histogram includes some that are double-counted. Table S4 contains the data used to generate Fig. 4.

Figure 4 Packages with optimization framework dependency.

Count of dependencies on an optimization related package, see “Optimization of R packages” section for complete list, by year a package was last updated. Packages with no stated dependency from their DESCRIPTION file for one of the specified packages are listed as “none.” In order to aid visual understanding of top dependencies, we limited display to those packages that had 14 or more dependent packages.

As the numbers from Metric 1 do not match the numbers of Metric 2, some additional exploration is necessary. In terms of total difference, there are 818 more packages using compiled code vs those with one of the searched for dependencies. There are 1,726 packages that do not list a dependency to one of the specified packages, but have a src directory for compiled code. There are 908 packages that list a dependency to one of the specified packages but do not have a src directory. See Tables S3 and S4 for more details.

Recommendations to Improve Quality and Performance

Software testing

Whenever software is written as part of a research project, careful consideration should be given to how to verify that the software performs the desired functionality and produces the desired output. As with bench science, software can often have unexpected and unintended results due to minor or even major problems during the implementation process. Software testing is a well-established component of any software development lifecycle (Atchison et al., 1968) and should also be a key component of research software. As shown previously, even among R software packages intended to be shared with and used by others, the majority of R packages (67–73% depending on metric) do not have tests that are made available with the package.

Various methodologies and strategies exist for software testing and validation as well as how to integrate software with a software development lifecycle. Some common testing strategies are no strategy, ad hoc testing (Agruss & Johnson, 2000), test driven development (TDD) (Beck & Gamma, 1998). There are also common project methodologies where testing fits into the project lifecycle; two common examples are the waterfall project management methodology, where testing is a major phase that occurs at a specific point in time, and the agile project management methodology (Beck et al., 2001), where there are many small iterations including testing. While a full discussion of various methods and strategies is beyond the scope of this article, three key concepts presented are: when to start testing, what to test, and how to test.

Key recommendations for when to test: Build tests before implementation.

Test after functionality has been implemented.

Discussion: One of the popular movements in recent years has been to develop tests first and then implement code to meet desired functionality, a strategy called TDD. While the TDD strategy has done much to improve the focus of the software engineering world on testing, some have found that it does not work with all development styles (Hansson, 2014; Sommerville, 2016), and others have reported that it does not increase developer productivity, reduce overall testing effort, nor improve code quality in comparison to other testing methodologies (Fucci et al., 2016). An approach that more closely matches the theoretically based software development cycle and flexible nature of research software is to create tests after a requirement or feature has been implemented (Osborne et al., 2014; Kanewala & Bieman, 2014). As developing comprehensive tests of software functionality can be a large burden to accrue at a single point in time, a more pragmatic approach is to alternate between developing new functionality and designing tests to validate new functionality. Similar to the agile software development strategy, a build/test cycle can allow for quick cycles of validated functionality that help to provide input into additional phases of the software lifecycle.

Key recommendations for what to test: Identify the most important or unique feature(s) of software being implemented. Software bugs are found to follow a Pareto or Zipfian distribution.

Test data and software configuration.

If performance is a key feature, build tests to evaluate performance.

Discussion: In an ideal world, any software developed would be accompanied by 100% test coverage validating all lines of code, all aspects of functionality, all input, and all interaction with other software. However, due to pressures of research, having time to build a perfect test suite is not realistic. A parsimonious application of the Pareto principle will go a long way towards improving overall software quality without adding to the testing burden. Large companies such as Microsoft have applied traditional scientific methods to the study of bugs and found that the Pareto principle matches reality: 20% of bugs cause 80% of problems; additionally a Zipfian distribution may apply as well: 1% of bugs cause 50% of all problems (Rooney, 2002).

To apply the Pareto principle to testing, spend some time in a thought experiment to determine answers to questions such as: What is the most important feature(s) of this software? If this software breaks, what is the most likely bad outcome? For computationally intensive components—how long should this take to run?

Once answers to these questions are known, the developer(s) should spend time designing tests to validate key features, avoiding major negatives, and ensuring software performs adequately. Optimization and performance recommendations are covered in the “Software Optimization” section. Part of the test design process should include how to “test” more than just the code. Some specific aspects of non-code tests include validation of approach and implementation choices with a mentor or colleague.

As a brief example of how to apply the aforementioned testing principles, we provide some information on testing steps followed during the pccc package development process. The first tests written were those that were manually developed and manually run as development progressed. Key test cases of this form are ideal candidates for inclusion in automated testing. The first tests were taking a known data set, running our process to identify how many of the input rows had complex chronic conditions, and then report on the total percentages found; this result was then compared with published values.

# read in HCUP KID 2009 Database kid9cols <- read_csv(“KID09_core_columns.csv”) kid9core <- read_fwf(“KID_2009_Core.ASC”,        fwf_positions(          start = kid9cols$start,          end = kid9cols$end,          col_names = tolower(kid9cols$name)),        col_types = paste(rep(“c”, nrow(kid9cols)),          collapse = “”)) # Output some summary information for manual inspection table(kid9core$year) dim(kid9core) n_distinct(kid9core$recnum) # Run process to identify complex chronic conditions kid_ccc <-    ccc(kid9core[, c(2, 24:48, 74:77, 106:120)],    id = recnum,    dx_cols = vars(starts_with(“dx”), starts_with(“ecode”)),    pc_cols = vars(starts_with(“pr”)),    icdv = 09) # Output results for manual inspection kid_ccc # Create summary statistics to compare to published values dplyr::summarize_at(kid_ccc, vars(-recnum), sum) %>% print.data.frame dplyr::summarize_at(kid_ccc, vars(-recnum), mean) %>% print.data.frame

For the pccc package there is a large set of ICD codes and code set patterns that are used to determine if an input record meets any complex chronic condition criteria. To validate the correct functioning of the software, we needed to validate the ICD code groupings were correct and were mutually exclusive (as appropriate). As pccc is a re-implementation of existing SAS and Stata code, we needed to validate that the codes from the previously developed and published software applications were identical and were performing as expected. Through a combination of manual review and automated comparison codes were checked to see if duplicates and overlaps existed. Any software dealing with input validation or having a large amount of built-in values used for key functionality should follow a similar data validation process.

As an example of configuration testing, here is a brief snippet of some of the code used to automatically find duplicates and codes that were already included as part of another code:

icds <- input.file(“../pccc_validation/icd10_codes_r.txt”) unlist(lapply(icds, function(i) {   tmp <- icds[icds != i]   output <- tmp[grepl(paste0(“^”, i, “.*”), tmp)]   # add the matched element into the output   if(length(output) != 0)     output <- c(i, output)   output }))

Key recommendations for how to test: Software developer develops unit tests.

Intended user of software should perform validation/acceptance tests.

Run all tests regularly.

Review key algorithms with domain experts.

Discussion: Most programming languages have a multitude of testing tools and frameworks available for assisting developers with the process of testing software. Due to the recurring patterns common across programming languages most languages have a SUnit (Wikipedia contributors, 2017a) derived testing tool, commonly referred to as an “xUnit” (Wikipedia contributors, 2017b) testing framework that focuses on validating individual units of code along with necessary input and output meet desired requirements. Based on software language used, unit tests may be at the class or function/procedure level. Some common xUnit style packages in R are RUnit and testthat. Unit tests should be automated and run regularly to ensure errors are caught and addressed quickly. For R, it is easy to integrate unit tests into the package build process, but other approaches such as post-commit hook in a version control system are also common.

In addition to unit tests, typically written by the developers of the software, users should perform acceptance tests, or high-level functionality tests that validate the software meets requirements. Due to the high-level nature and subjective focus of acceptance tests, they are often manually performed and may not follow a regimented series of steps. Careful documentation of how a user will actually use software, referred to as user stories, are translated into step by step tests that a human follows to validate the software works as expected. A few examples of acceptance testing tools that primarily focus GUI aspects of software are: Selenium (Selenium Contributors, 2018), Microfocus Unified Functional Testing (formely known as HP’s QuickTest Professional) (Micro Focus, 2018), and Ranorex (Ranorex GmbH, 2018). As research focused software often does not have a GUI, one aide to manual testing processes is for developers of the software or expert users to create a full step by step example via an R Markdown (Allaire et al., 2018b; Xie, Allaire & Grolemund, 2018) notebook demonstrating use of the software followed by either manually or automatic validation that the expected end result is correct.

In addition to the tool-based approaches already mentioned, other harder to test items such as algorithms and solution approach should be scrutinized as well. While automated tests can validate mathematical operations or other logic steps are correct, they cannot verify that the approach or assumptions implied through software operations are correct. This level of testing can be done through code review and design review sessions with others who have knowledge of the domain or a related domain.

During development of the pccc package, after the initial tests shown in previous sections, further thought went into how the specifics of the desired functionality should perform. Unit tests were developed to validate core functionality. We also spent time thinking about how the software might behave if the input data was incorrect or if parameters were not specified correctly. If an issue is discovered at this point, a common pattern is to create a test case for discovered bugs that are fixed—this ensures that a re-occurrence, known as a “regression” to software engineers, of this error does not happen again. In the case of pccc, developers expected large input comprised of many observations with many variables. When a tester accidentally just passed 1 observation with many variables, the program crashed. The problem was discovered to be due to the flexible nature of the sapply() function returning different data types based on input.

The original code from ccc.R: # check if call didn’t specify specific diagnosis columns if (!missing(dx_cols)) {    # assume columns are referenced by ‘dx_cols’    dxmat <- sapply(dplyr::select(      data, !!dplyr::enquo(dx_cols)), as.character)    # create empty matrix if necessary    if(! is.matrix(dxmat)) {      dxmat <- as.matrix(dxmat)    } } else {    dxmat <- matrix(“”, nrow = nrow(data)) }

The new code:

if (!missing(dx_cols)) {    dxmat <- as.matrix(dplyr::mutate_all(      dplyr::select(        data, !!dplyr::enquo(dx_cols)),      as.character)) } else {    dxmat <- matrix(“”, nrow = nrow(data)) }

One of the tests written to verify the problem didn’t reoccur:

# Due to previous use of sapply in ccc.R, this would fail test_that(paste(“1 patient with multiple rows of no diagnosis”,           “data–should have all CCCs as FALSE”), {   expect_true(all(ccc(dplyr::data_frame(     id = ‘a’,     dx1 = NA,     dx2 = NA),    dx_cols = dplyr::starts_with(“dx”),    icdv = code) == 0))   } )

Testing Anti-Patterns: While the above guidance should help researchers know the basics of testing, it does not cover in detail what not to do. An excellent collection of testing anti-patterns can be found at (Moilanen, 2014; Carr, 2015; Stack Overflow Contributors, 2017). Some key problems that novices experience when learning how to test software are: Interdependent tests—Interdependent tests can cause multiple test failures. When a failure in an early test case breaks a later test, it can cause difficulty in resolution and remediation.

Testing application performance—While testing execution timing or software performance is a good idea and is covered more in the “Software Optimization” section, creating an automated test to perform this is difficult and does not carry over well from one machine to another.

Slow running tests—As much as possible, tests should be automated but still run quickly. If the testing process takes too long consider refactoring tests or evaluating the performance of the software being tested.

Only test correct input—A common problem in testing is to only validate expected inputs and desired behavior. Make sure tests cover invalid input, exceptions, and similar items.

Software optimization

Getting software to run in a reasonable amount of time is always a key consideration when working with large datasets. A mathematical understanding of software algorithms is usually a key component of software engineering curricula, but not widely covered in other disciplines. Additionally, while software engineering texts and curricula highlight the importance of testing for non-functional requirements such as performance (Sommerville, 2015), they often fail to provide details on how best to evaluate software performance or how to plan for performance during the various phases of software lifecycle.

The survey of R packages at the beginning of this work indicates that approximately 75% of packages do not use optimization related packages nor compiled code to improve performance. While the survey of R packages is not evidence of non-optimization of packages in CRAN, computational researchers can should carefully consider performance aspects of their software before declaring it complete. This section will provide a starting point for additional study, research, and experimentation. The Python Foundation’s Python language wiki provides excellent high-level advice (Python Wiki Contributors, 2018) to follow before spending too much time in optimization: First get the software working correctly, test to see if it is correct, profile the application if it is slow, and lastly optimize based on the results of code profiling. If necessary, repeat multiple cycles of testing, profiling, and optimization phases. The key aspects of software optimization discussed in this are: identify a performance target, understanding and applying Big O notation, and the use code profiling and benchmarking tools.

Key recommendations for identifying and validating performance targets: Identify functional and non-functional requirements of the software being developed.

If software performance is key to the software requirements, develop repeatable tests to evaluate performance.

Discussion: The first step to software optimization is to understand the functional and non-functional requirements of the software being built. Based on expected input, output, and platform the software will be run on, one can make a decision as to what is good enough for the software being developed. A pragmatic approach is best—do not spend time optimizing if it does not add value. Once the functional requirements have been correctly implemented and validated, a decision point is reached: decide if the software is slow and in need of evaluation and optimization. While this may seem a trivial and unnecessary step, it should not be overlooked; a careful evaluation of costs versus benefit from an optimization effort should be evaluated before moving forward. Some methods for gathering the performance target are through an evaluation of other similar software, interdependencies of the software and its interaction with other systems, and discussion with other experts in the field.

Once a performance target has been identified, development of tests for performance can begin. While performance testing is often considered an anti-pattern of testing (Moilanen, 2014) some repeatable tests should be created to track performance as development progresses. Often a “stress test” or a test with greater than expected input/usage is the best way to do this. A good target is to check an order of magnitude larger input than expected. This type of testing can provide valuable insight into the performance characteristics of the software as well unearth potentials for failure due to unexpected load (Sommerville, 2015).

Here is an example of performance validation testing that can also serve as a basic reproducibility test calling the main function from pccc using the microbenchmark package (one could also use bench, benchr, or other similar R packages).

library(pccc) rm(list=ls()) gc() icd10_large <-   feather::read_feather(    “../icd_file_generator/icd10_sample_large.feather” ) library(microbenchmark) microbenchmark( ccc(icd10_large[1:10000, c(1:45)], # get id, dx, and pc columns           id = id,           dx_cols = dplyr::starts_with(“dx”),           pc_cols = dplyr::starts_with(“pc”),           icdv = 10), times = 10) Unit: seconds expr   min    lq   mean  median    uq    max  neval ccc 2.857625 2.908964 2.959805 2.920408 3.023602 3.119937   10

Results are from a system with 3.1 GHz Intel Core i7, 16 GB 2133 MHz LPDDR3, PCI-Express SSD, running macOS 10.12.6 and R version 3.5.1 (2018-07-02).

As software runs can differ significantly from one to the next due to other software running on the test system, a good starting point is to run the same test 10 times (rather than the microbenchmark default of 100 due to this being a longer running process) and record the mean run time. microbenchmark also shows median, lower and upper quartiles, min, and max run times. The actual ccc() call specifics are un-important; the key is to test the main features of your software in a repeatable fashion and watch for performance changes over time. These metrics can help to identify if a test was valid and indicate a need for retesting; that is, a large interquartile range may indicate not enough tests were run or some aspect of environment is causing performance variations. Software benchmarking is highly system specific in that changing OS version, R version, R dependent package version, compiler version (if compiled code involved), or hardware may change the results. As long as all tests are run the same on the same system with the same software, one can compare timings as development progresses.

Lastly, although the example above is focused on runtime, it can be beneficial to also identify targets for disk space used and memory required to complete all desired tasks. As an example, tools such as bench and profvis demonstrated in our “Code Profiling/Benchmarking” section as well as object.size() from core R can give developers insight into memory allocation and usage. There are many resources beyond this work that can provide guidance on how to minimize RAM and disk resources (Kane, Emerson & Weston, 2013; Wickham, 2014b; Wickham et al., 2016; Klik, Collet & Facebook, 2018).

Key recommendations for identifying upper bound on performance: Big O notation allows the comparison of theoretical performance of different algorithms.

Evaluate how many times blocks of code will run as input approaches infinity.

Loops inside loops are very slow as input approaches infinity.

Discussion: Big O notation is a method for mathematically determining the upper bound on performance of a block of code without consideration for language and hardware specifics. Although performance can be evaluated in terms of storage or run time, most examples and comparisons focus on run time. However, when working with large datasets, memory usage and disk usage can be of equal or higher importance than run. Big O notation is reported in terms of input (usually denoted as n) and allows one to quickly compare theoretical performance of different algorithms.

The basic steps for evaluating the upper bound of performance of a block of software code is to evaluate what code will run as n approaches infinity. Items that are constant time (regardless of if they run once or x times independent of input) are reduced down to O(1). The key factors that contribute to Big O are loops—a single for loop or similar construct through recursion that runs once for all n is O(n); a nested for loop would be O(n2). When calculating Big O for a code block, function, or software system, lower order terms are ignored, and just the largest Big O notation is used; for example, if a code block is O(1) + O(n) + O(n3) it would be denoted as O(n3).

Despite the value of understanding the theoretical upper bound of software in an ideal situation, there are many difficulties that arise during implementation that can make Big O difficult to calculate and which could make a large Big O faster than a small Big O under actual input conditions. Some key takeaways to temper a mathematical evaluation of Big O are: Constants matter when choosing an algorithm—for example, if one algorithm is O(56n2), there exists some n where O(n3) is faster.

Average or best case run time might be more relevant.

Big O evaluation of algorithms in high level languages is often hard to quantify.

For additional details on Big O notation, see the excellent and broadly understandable introduction to Big O notation (Abrahms, 2016).

Key recommendations for profiling and benchmarking: Profile code to find bottlenecks.

Modify code to address largest items from profiling.

Run tests to make sure functionality isn’t affected.

Repeat process if gains are made and additional performance improvements are necessary.

Discussion: As discussed throughout this section, optimization is a key aspect of software development, especially with respect to large datasets. Although identification of performance targets and a mathematical analysis of algorithms are important steps, the final result must be tested and verified. The only way to know if your software will perform adequately under ideal (and non-ideal) circumstances is to use benchmarking and code profiling tools. Code profilers show how a software behaves and what functions are being called while benchmarking tools generally focus on just execution time—though some tools combine both profiling and benchmarking. In R, some of the common tools are bench, benchr, microbenchmark, tictoc, Rprof (R Core Team, 2018), proftools (Tierney & Jarjour, 2016), and profvis.

If, after implementation has been completed, the software functions correctly, and performance targets have not been met, look to optimize your code. Follow an iterative process of profiling to find bottlenecks, making software adjustments, testing small sections with benchmarking and then repeating the process with overall profiling again. If at any point in the process you discover that due to input size, functional requirements, hardware limitations, or software dependencies you cannot make a significant impact to performance, consider stopping further optimization efforts (Burns, 2012).

As with software testing and software bugs, the Pareto principle applies, though some put the balance between code and execution time is closer to 90% of time is in 10% of the code or even as high as 99% in 1% (Xochellis, 2010; Bird, 2013). Identify the biggest bottlenecks via code profiling and focus only on the top issues first. As an example of how to perform code profiling and benchmarking in R, do the following:

First, use profvis to identify the location with the largest execution time:

library(pccc) icd10_large <- feather::read_feather(“icd10_sample_large.feather”) profvis::profvis({ccc(icd10_large[1:10000,],              id = id,              dx_cols = dplyr::starts_with(“dx”),              pc_cols = dplyr::starts_with(“pc”),              icdv = 10)}, torture = 100)

In Fig. 5 you can see a visual depiction of memory allocation, known as a “Flame Graph,” as well as execution time and call stack. By clicking on each item in the stack you will be taken directly to the relevant source code and can see which portions of the code take the most time or memory allocations. Figure 6 is a depiction of the data view which shows just the memory changes, execution time, and source file.

Figure 5 Profvis flame graph.

Visual depiction of memory allocation/deallocation, execution time, and call stack.

Figure 6 Profvis data chart.

Table view of memory allocation/deallocation, execution time, and call stack.

Once the bottleneck has been identified, if possible extract that code to a single function or line that can be run repeatedly with a library such as microbenchmark or tictoc to see if a small change either improves or degrades performance. Test frequently and make sure to compare against previous versions. You may find that something you thought would improve performance degrades performance. As a first step we recommend running tictoc to get general timings such as the following:

library(tictoc) tic(“timing: r version”) out <- dplyr::bind_cols(ids, ccc_mat_r(dxmat, pcmat, icdv)) toc() tic(“timing: c++ version”) dplyr::bind_cols(ids, ccc_mat_rcpp(dxmat, pcmat, icdv)) toc() timing: r version: 37.089 sec elapsed timing: c++ version: 5.087 sec elapsed

As with previous timings, while we’re showing pccc calls, any custom function of block of code you have can be compared against an alternative version to see which performs better. The above blocks of code call the core functionality of the pccc package—one implemented all in R, the other with C++ for the matrix processing and string matching components; see sourcecode available at https://github.com/magic-lantern/pccc/blob/no_cpp/R/ccc.R for full listing.

After starting with high level timings, next run benchmarks on specific sections of code such as in this example comparing importing a package vs using the package reference operator using bench:

library(bench) set.seed(42) bench::mark(   package_ref <- lapply(medium_input, function(i) {    if(any(stringi::stri_startswith_fixed(i, ‘S’),na.rm = TRUE))      return(1L)    else      return(0L) })) # A tibble: 1 × 14   expression   min  mean  median   max  `itr/sec`   mem_alloc     <chr>   <bch>  <bch>  <bch:>  <bch>   <dbl>   <bch:byt> 1 package_r…  547ms  547ms  547ms  547ms    1.83    17.9MB library(stringi) bench::mark(    direct_ref <- lapply(medium_input, function(i) {    if(any(stri_startswith_fixed(i, ‘S’),na.rm = TRUE))      return(1L)    else      return(0L) })) # A tibble: 1 × 14     expression   min   mean   median   max   `itr/sec`   mem_alloc    <chr>    <bch>  <bch>   <bch:>  <bch>    <dbl>  <bch:byt>   1 direct_re…  271ms  274ms   274ms  277ms    3.65    17.9MB

The above test was run on a virtual machine running Ubuntu 16.04.5 LTS using R 3.4.4.

One benefit of bench::mark over microbenchmark is that bench reports memory allocations as well as timings, similar to data shown in profvis. Through benchmarking we found that for some systems/configurations the use of the “::” operator, as opposed to importing a package, worsened performance noticeably. Also widely known (Gillespie & Lovelace, 2017) and found to be applicable here is that the use of matrices are preferred for performance reasons over data.frames or tibbles. Matrices do have different functionality, which can require some re-work when converting from one to another. For example, a matrix can only contain 1 data type such as character or numeric; data.frames and tibbles support shortcut notations such as mydf$colname. Another key point found is that an “env” with no parent environment is significantly faster (up to 50x) than one with a parent env. In the end, optimization efforts resulted in reducing run time by 80%.

One limitation with R profiling tools is that if the code to be profiled executes C++ code, you will get no visibility into what is happening once the switch from R to C++ has occurred. As shown in Fig. 7, visibility into timing and memory allocation stops at the .Call() function. In order to profile C++ code, you need to use non-R specific tools such as XCode on macOS or gprof on non-macOS Unix-based operating system (OS). See “R_with_C++_profiling.md” in our source code repository for some guidance on this topic.

Figure 7 Profvis flame graph .Call().

Visual depiction of memory allocation/deallocation, execution time, and call stack; note the limitations in detail at the .Call() function where custom compiled code is called.

Some general lessons learned from profiling and benchmarking: “Beware the dangers of premature optimization of your code. Your first duty is to create clear, correct code.” (Knuth, 1974; Burns, 2012) Never optimize before you actually know what is taking all the time/memory/space with your software. Different compilers and core language updates often will change or reverse what experience has previously indicated as sources of slowness. Always benchmark and profile before making a change.

Start development with a high-level programming language first—Developer/Researcher time is more valuable than CPU/GPU time. Choose the language that allows the developer/researcher to rapidly implement the desired functionality rather than selecting a language/framework based on artificial benchmarks (Kelleher & Pausch, 2005; Jones & Bonsignour, 2011).

Software timing is highly OS, compiler, and system configuration specific. What improves results greatly on one machine and configuration may actually slow performance on another machine. Once you decided to put effort into optimization, make sure you test on a range of realistic configurations before deciding that an “improvement” is beneficial (Hyde, 2009).

If you’ve exhausted your options with your chosen high-level language, C++ is usually the best option for further optimization. For an excellent introduction to combining C++ with R via the library Rcpp, see (Eddelbuettel & Balamuta, 2017).

For some additional information on R optimization, see (Wickham, 2014b; Robinson, 2017).

Conclusion

Researchers frequently develop software to automate tasks and speed the pace of research. Unfortunately, researchers are rarely trained in software engineering principles necessary to develop robust, validated, and performant software. Software maintenance is an often overlooked and underestimated aspect in the lifecycle of any software product. Software engineering principles and tooling place special focus on the processes around designing, building, and maintaining software. In this paper, the key topics of software testing and software optimization have been discussed along with some analysis of existing software packages in the R language. Our analysis showed that the majority of R packages have neither unit testing nor evidence of optimization available with normally distributed source code. Through self-education on unit testing and optimization, any computational or other researcher can pick up the key principles of software engineering that will enable them to spend less time troubleshooting software and more time doing the research they enjoy.

Supplemental Information

Supplemental Information 1 Packages by year updated with non-empty testing directory.

“All” column summarizes data from years from to 2005 up through 2018.

Click here for additional data file.

Supplemental Information 2 Packages by year updated and testing framework dependency.

Only non-zero percentages shown; “All” column summarizes data from years from to 2005 up through 2018.

Click here for additional data file.

Supplemental Information 3 Packages by year updated with non-empty src directory.

“All” column summarizes data from years from to 2005 up through 2018.

Click here for additional data file.

Supplemental Information 4 Packages by year updated and optimization framework dependency.

Only non-zero percentages shown; “All” column summarizes data from years from to 2005 up through 2018.

Click here for additional data file.

Additional Information and Declarations

Competing Interests

Author Contributions

Data Availability

The authors declare that they have no competing interests.

Seth Russell conceived and designed the experiments, performed the experiments, analyzed the data, contributed reagents/materials/analysis tools, prepared figures and/or tables, performed the computation work, authored or reviewed drafts of the paper, approved the final draft.

Tellen D. Bennett conceived and designed the experiments, authored or reviewed drafts of the paper, approved the final draft.

Debashis Ghosh conceived and designed the experiments, authored or reviewed drafts of the paper, approved the final draft.

The following information was supplied regarding data availability:

Source code, images, generated data for R package analysis are available at: https://github.com/magic-lantern/SoftwareEngineeringPrinciples.

Materials from paper including figures, references, and text are available at: https://github.com/magic-lantern/SoftwareEngineeringPrinciples/tree/master/paper.

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
