# Peer review of "Software engineering principles to improve quality and performance of R software"

_PeerJ Computer Science, doi:10.7717/peerj-cs.175_

## Round 0.1 · original submission · Major Revisions

As you can check, both reviewers have several interesting comments and corrections to the paper. Please address both of them.

·

Basic reporting

I think for the sake of saving people the need for a dictionary, "autodictacts" should be replaced by "self-taught individuals" and "codified through" should be replaced by "constructed using" or something like that.
Line 69 is missing a full stop.
Line 129 is missing its last word (years)
Line 140 has a misformatted citation (jpresse)
Line 166 has "overtime" where it should say "over time"
Line 167 should say "percentage" and not "percent"
Line 483 has a misformed citation
Other than this, the English used is clear and professional and the citations provide nice
background/context.

The raw data is shared.

The article is self-contained with relevant results to hypotheses.

Experimental design

The research question is well defined, relevant & meaningful. It is stated how this research fills an identified knowledge gap.

Rigorous investigation is performed to a high technical and ethical standard.

Methods are described with sufficient detail and info te replicate.

Validity of the findings

The findings are valid and the authors have gone to considerable and sufficient effort to ensure that the data is robust.

The conclusions are well stated.

Additional comments

I find the article interesting and extremely relevant. In particular I welcome the improved analysis of unit testing in CRAN packages relative to what I did myself a few years ago.

I find the examples of unit testing using the pccc package a little difficult to understand. Sure, they can be understood with some effort, but I think there must be an easier example out there which would permit a more concise code listing. Perhaps the glue package? I know pccc is a niche example and you're probably trying to encourage niche developers to test their packages too, but I think for the sake of example, simpler is better.

The study of packages employing optimization is very interesting. However, with testing I think it's easy to say if a package isn't tested, it should be, but a package without obvious optimization attempts could just be very well written (using only vectorized code from other packages) and hence not need more explicit optimizations of its own. I think this should be stated: a package without obvious optimization isn't necessarily in need of optimization. Having said this, figure 3 is interesting, giving the change in explicit optimization efforts over the years.

In figure 1, I think the "No" colour should be on the bottom and in particular in fig 4, "None" should be on the bottom, not in the middle. "None" should maybe also be given a striking colour to show that it's different to the other results.
It should be made clear in the caption of fig 4 what the n>14 is about.

Consider using bench::mark instead of microbenchmark (I read recently that bench is better and the tidyverse people are pushing it now).

Thanks for a very nice article.

·

Basic reporting

- Professional structure (but I have asked for some reorganization).
- References provided are good, but more are needed to substantiate specific claims (see general discussion below).

Experimental design

- Data collection and analysis is good.

Validity of the findings

- Empirical results on testing and performance optimization seem trustworthy.
- High-level discussion of rules to follow only partly backed up by empirical evidence (see general discussion).

Additional comments

General:

I feel the authors are trying to:

1. Introduce readers to key ideas and tools in software testing.

2. Survey the current state of software testing for R packages.

3. Do both of the above all over again for performance optimization.

The empirical study of how many packages on CRAN do testing and optimization was the most interesting part of the paper for me, but I acknowledge that I'm not representative of likely readers of this paper. I think that the overview of testing the authors give in lines 180-262 is not detailed enough to serve as a tutorial for people who aren't already familiar with the topic, while being too long for people who _are_ already familiar with it. (I feel the same way about the discussion of big-O notation starting at L440.) I therefore recommend that the authors:

1. Put the empirical material on testing and optimization in one section near the front of the paper to show that there is significant room for improvement.

2. Replace the paragraph-length high-level advice on how to test and tune with bulleted lists of rules, each having pointers to longer-form discussions. This will help experienced readers (who will nod at the lists of rules), while also helping newcomers operationalize those rules (which I think they would struggle to do based on the current brief explanations).

3. Expand the examples to show specific applications of the general rules. For example, I would like to see the performance results for the PCCC code on line 413 and following, and then see what changes the authors made to the code to speed it up, and a second set of performance figures. Similarly, in the testing example starting on line 301, I do not know what the bug was that the test found, or how the bug was detected before the test was written and its detection then translated into a test.
* * *
Specific:

28: "we show that reproducible and replicable software tests are not available" -> as written, this is a very strong claim.

39: Introduction of software engineering seems disconnected with preceding material.

69: missing "." between "maintenance" and "Software maintenance"

71: "chief factor" -> isn't people's time the chief factor in _all_ maintenance, not just that for statistical software?

76: "As research scientists tend to think..." -> Are they wrong? I.e., if I assert that the majority of software written by researchers exists to solve one-off problems, rather than to be used repeatedly, is there data to show that I'm wrong?

81: Here and elsewhere, I worry that "best practices" is not validated. There is, for example, no published research showing that the use of version control makes people more productive. (Believe me, I've looked.) I think the authors need to present evidence that various practices actually improve productivity and/or reliability before calling them "best".

102: Do the authors have data showing that commercial software and/or open source software are tested any more frequently? (I've seen a _lot_ of projects on GitHub that don't have any tests...)

140: What is "jpreese"?

149: I'm unclear what is meant by "Grep for...directories" - are the authors working from textual manifests of projects, or are they using "grep" as a synonym for "search for"?

160: should "updated" be "most recently updated"?

166: "over time" rather than "overtime"

172: a table might be a better way to display this summary of "have X but not Y".

184: There is also growing evidence that TDD doesn't actually make developers more productive (see e.g. http://people.brunel.ac.uk/~csstmms/FucciEtAl_ESEM2016.pdf).

185: "A better approach..." Better by what criteria, and what data can the authors cite to support this contention?

194: "In an ideal world...100% test coverage" Do the authors mean line coverage, statement coverage, branch coverage, combinatorial coverage, ...?

208: "Once answers to these questions are known..." I have found that giving high-level advice like this only frustrates most scientists, because they don't know how to operationalize it. Can the authors point at concrete examples of how to translate these general rules into specific decisions, priorities, and/or tests for specific software packages?

226: "In addition to unit tests, users should perform..." I believe the authors mean "developers", not "users" (but could be wrong). I also think that this statement trips over an important distinction between testing what-if contingencies for software tools (which developers of packages should do), and testing specific users of those tools and their inputs for particular analyses (which analysts doing particular analyses should do). I think that repeatable sanity checks make sense for the latter, but that doesn't necessarily mean use of unit testing frameworks.

233: A notebook demonstrating the use of the software is _not_ the same thing as an acceptance test, though the automatic validation of the expected end result may be.

238: Radcliffe et al have developed a nice framework for thinking about the ways in which data analysis can go wrong, which is summarized in the figure in http://stochasticsolutions.com/pdf/TDDA-One-Pager.pdf. It seems that most of the discussion in this paper is focusing on Step 2 of that model's 5-step process - if so, the authors may wish to cite that model and make this explicit, and if not, expand their recommendations for other phases.

301-302: some odd indentation.

430: "run 10 timess and find the mean" No - this can easily give a misleadingly high performance result because of caching effects.

437: "There are many optimizations that can be considered..." This is the kind of advice that my students used to find frustrating, because it tells readers that something exists without telling them what it is or where to find it.

560: the summary of possible optimizations is very condensed - can the authors point at resources that have lengthier and more detailed coverage? (There are several guides to high performance R...)

---

## Round 0.2 · accepted · Accept

This new version improves considerably the previous one. Now the paper is ready to be accepted. Congratulations.

Reviewer 3 ·

Basic reporting

See the general comments.

Experimental design

See the general comments.

Validity of the findings

The discussion and conclusions are well stated, they are limited to the results obtained.

Additional comments

The paper is interesting and motivating. The main goal and motivation of the paper are clear. Nowadays, the development of scientific software needs a guide to optimize and test the code and, therefore, to improve software maintenance.

Generally speaking, I consider the work has been significantly improved. The authors have effectively addressed all the comments raised in the previous version.

I think that this work can be accepted for publication.